# Use of Printed Sensors to Measure Strain in Rolling Bearings under Isolated Boundary Conditions

Marcel Bartz [1,*], Felix Häußler [2], Fabian Halmos [1], Markus Ankenbrand [2], Michael Jüttner [1], Jewgeni Roudenko [3], Sven Wirsching [1], Marcus Reichenberger [3], Jörg Franke [2] and Sandro Wartzack [1]

1 Department of Mechanical Engineering, Friedrich-Alexander-Universität Erlangen-Nürnberg (FAU), Engineering Design, Martensstraße 9, 91058 Erlangen, Germany; wartzack@mfk.fau.de (S.W.)

2 Department of Mechanical Engineering, Friedrich-Alexander-Universität Erlangen-Nürnberg (FAU), Institute for Factory Automation and Production Systems, Egerlandstraße 7, 91058 Erlangen, Germany; joerg.franke@faps.fau.de (J.F.)

3 Institute for Chemistry, Materials- and Product Development (OHM-CMP), Technische Hochschule Nürnberg Georg Simon Ohm, Keßlerplatz 12, 90489 Nuremberg, Germany; jewgeni.roudenko@th-nuernberg.de (J.R.)

* Correspondence: bartz@mfk.fau.de

**Abstract:** The knowledge of the operating conditions in rolling bearings in technical applications offers many advantages, for example, to ensure a safe operation and to save resources and costs with the help of condition monitoring and predictive maintenance procedures. In many cases, it is difficult to implement sensors to measure the operating conditions of the rolling bearing, for reasons such as inaccessibility of the mounting position or non-compliance with installation space neutrality, which influences the sensor on the measuring point. Printed sensors using a digital deposition process, which can be used in very narrow design spaces, offer advantages in this respect. So far, these sensors have not been established in rolling bearings, so there is potential for technical application. This paper discusses the fundamental advantages and disadvantages as well as the challenges of the application, and it demonstrates the feasibility under isolated boundary conditions by applying a printed strain gauge sensor to the outer ring of a cylindrical roller bearing NU210 in an experimental setup to measure the strain under load. In this setup, the outer ring is deformed by 2 mm under an increasing radial load using a hydraulic press, and the strain is measured. Both a commercial reference sensor and a FE-simulation are used to validate the measurement. The results show that an implementation using printed sensors as a strain gauge works successfully. The resulting challenges, such as measuring strain gradients and printing on curved surfaces, are finally evaluated, and an outlook for further work is given.

**Keywords:** rolling bearings; sensor-integrated machine elements; strain measurement; printed strain gauges; printed sensors

## 1. Introduction

### 1.1. Motivation and State-of-the-Art

Rolling bearings are one of the most widely used machine elements in technical systems and have become essential for enabling rotating motion in modern applications. For example, rolling bearings are used in the fields of mobility and manufacturing technology as well as in energy generation. Applications include electromobility, rail vehicles and wind turbines. Research and development in the field of rolling bearing technology therefore makes an important contribution to the transformation of mobility and the development of sustainable energy resources [1].

A trend that also offers great potential for the further development of rolling bearings is the ongoing digitalization [2], in which mechanical components and systems [3] are being developed into intelligent, mechatronic systems [4], combining sensors, actuators and information processing with predictive, networked and self-acting components. Great

importance is attached to the reliable acquisition of process and condition data using suitable sensors, on the basis of which decisions are made for the system [5,6].

The important process and condition data of rolling bearings or systems in which rolling bearings are installed include the parameters of rotational speed of the rotating ring as well as the rolling elements and cage; load distribution and deformation [7]; temperature [8] of the bearing components as well as the lubricant, the lubricant film height and vibrations [9]. The measurement of load and temperature distributions, deformations, rotational frequencies or vibrations enables, for example, condition-based maintenance and servicing [10]. In addition, measurement with sensors allows the control of the entire system in which the rolling bearings are installed by identifying and avoiding critical operating conditions and increasing the accuracy of fatigue life estimates by reducing uncertainties in load assumptions.

While some parameters, such as rotational speed and temperature, can be measured directly and easily, others can only be measured indirectly or with great difficulty. One important parameter that requires increased measurement effort is the deformation of the bearing rings in the vicinity of the load zone. One reason for this is the poor accessibility of the relevant measuring points as well as the challenging operating conditions at the measuring point due to high temperatures and over-rolling of the rolling elements. Furthermore, measurements should not influence the function and size of the bearing, so they should be as neutral as possible in terms of design space. However, deformation is an important parameter for determining the stresses acting in the material and thus providing information on load distribution and bearing life.

Established approaches to extending the functions of rolling bearings with sensors, for example [11,12], are aimed at mounting sensor modules (sensor-carrying machine elements). These approaches rely on cable-based power supply and data communication and therefore cannot be used without changing the surrounding system. There are also approaches based on the electrical impedance method for load and condition monitoring [13], which, however, are to be assigned to the sensor-based machine elements and can have a direct effect on the operating characteristics and service life of the bearing due to the passage of current [14]. In contrast, the advantage of sensor-integrated machine elements is that they are design space neutral and ideally have no negative influence on the bearing functions [6]. The parameters to be measured are directly related to the primary mechanical function of the rolling bearing.

For measuring the deformation of rolling bearing rings caused by mechanical and thermal stresses, resistive, capacitive, inductive, optical and piezoelectric sensor principles can basically be used. Nowadays, sensors are predominantly manufactured using microsystem technology methods and processes, which are extended using micromechanical methods such as 3D shaping [15]. In addition to this classical manufacturing process, it is also possible to manufacture elementary sensors using printing technology. Strain gauges, for example, can be manufactured using screen printing in thick-film technology, whereby greater sensitivity can be achieved than with classic metal-foil-based strain gauges [16–18]. However, when applying metal-foil-based strain gauges, additional mounting errors can falsify the measurement result [19]. High-precision and 3D-capable printing technologies, such as aerosol-based or inkjet processes offer potential for improvement here, allowing sensor functions to be implemented directly on the surfaces of machine elements, largely independent of the design space. The printed sensor structures consist of a meandering conducting path and thus function like conventional strain gauges. The utilization of printed electronics allows the creation of sensors on free-form surfaces with different materials and flexible layouts. Initial implementations of strain gauges and temperature sensors using digital printing processes are currently at the cutting edge of research and promise high accuracy and robustness [20–22].

The application of strain-gauge-printed sensors using the inkjet or the aerosol-based printing method to measure deformations on mechanical components has already been demonstrated [21,23]. However, the potential for measuring parameters in rolling bearings

has not yet been investigated. Therefore, the aim of this paper is to investigate the potential of the aerosol-based method for application to rolling bearings.

### 1.2. Research Objective of This Paper

The research objective of this paper is to estimate the potential of printed sensors as a strain gauge for rolling bearing deformation measurements. In the real application of a rolling bearing in a technical environment, the deformation of the bearing (especially of the inner and outer ring) is subject to different influences, e.g., varying dynamic loads, temperature influences and others. In order to investigate the potential of printed sensors for measuring the deformation of rolling bearings, this paper first examines the rolling bearing under isolated boundary conditions, i.e., in a laboratory environment with an externally applied load. The reason for this is that even under isolated conditions, there are enough challenges to investigate and discuss.

One of the research objectives in this paper is the positioning of the sensors in order to ensure neutrality of the installation space on the one hand and finding a position where relevant deformations occur on the other hand. In addition to the challenge of finding the optimum position, it is also necessary to ensure the printability of the sensor at this position. A rolling bearing consists mainly of curved surfaces, supplemented by ribs, radial sections and other non-planar geometries. While printed sensors on flat surfaces—e.g., tension rods—pose fewer challenges in terms of printability and also the validation of deformation measurements, in the case of curved surfaces, this is not a trivial issue. Therefore, the printability of the sensors on the curved rolling bearing surfaces represents another research objective of this paper.

Due to the challenges mentioned above, a simple isolated system is chosen for investigation in this paper, namely the outer ring of a cylindrical roller bearing subjected to an external load. This work is divided into the following parts: First, the investigation of the optimal location and printing of the sensors is carried out. Then, after applying a load and measuring the deformation, the experimental results are validated against expected strain values through FE-simulation calculations and a comparison of the measurement data with a (non-printed) reference sensor. Details of the positioning, printing and post-processing of the sensors, the experimental setup and the FE-simulation are described in the following Materials and Methods section.

In summary, the final goal of this paper is to demonstrate a validated application of printed sensors as strain gauges on a rolling bearing under isolated boundary conditions (i.e., not in real-world operation) in order to answer the questions raised by the above-mentioned challenges and to provide preliminary work for the use of printed sensor technology under real operating conditions.

## 2. Materials and Methods

In the following, the advantages and disadvantages of different printing positions on the rolling bearing are discussed; the process before printing as well as experimental and simulative setups are described.

### 2.1. Possible Positions for Sensor Mounting and Advantages and Disadvantages

There are various positions on a rolling bearing where a strain measurement could be of interest. Figure 1 shows potential positions where deformation can be used to make different conclusions, e.g., about the load distribution.

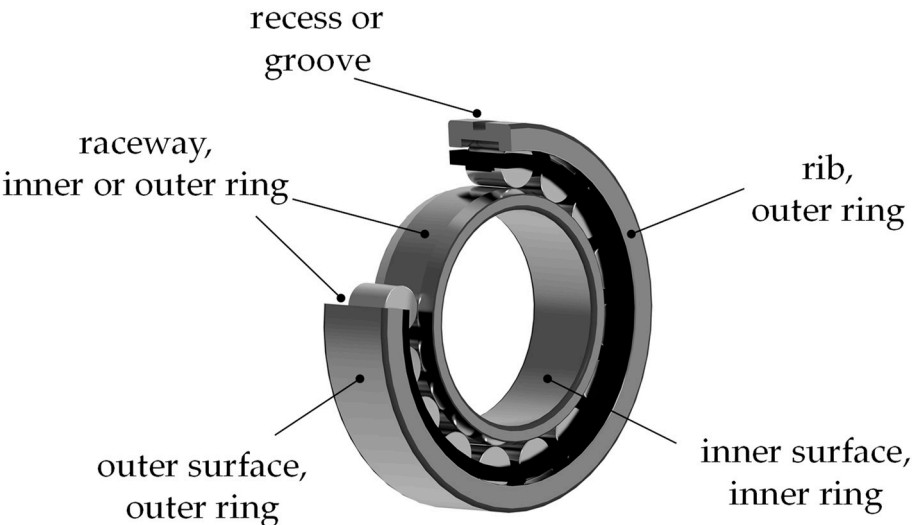

**Figure 1.** Potential positions for attaching sensors to measure strain values on a cylindrical roller bearing.

For a rolling bearing, e.g., a cylindrical roller bearing as shown in Figure 1, both the strain at the outer ring as well as at the inner ring can be of interest. From a mechanical point of view, the positioning of a strain gauge sensor depends, on one hand, on whether the bearing is loaded radially, axially or in combination, and on the other hand, on which of the two rings is rotating and how the load is distributed during operation. Depending on these operating conditions, the strain of the outer or inner ring, or both, can be interesting. To measure the strain of the rings, sensors can be mounted on the ribs or on the outer or inner surfaces of the rings. The most suitable positions are those that have the least effect on the operating behavior of the rolling bearing. For example, positioning sensors on the raceway would be interesting in terms of the measured data as they could be close to or within the load zone. However, they could influence the operating behavior and would quickly wear out during operation due to the strong over-rolling of the rolling elements. Another possibility for integrating sensors in a rolling bearing at interesting positions while minimizing their exposure to disturbing influences is to make a recess or groove in the rolling bearing ring, as shown for example in the upper section of Figure 1. While this option could be advantageous for sensor positioning, it involves additional manufacturing effort and may compromise the mechanical integrity of the rolling bearing rings and their function.

From the description above, it is clear that different sensor positions have certain advantages and disadvantages in terms of measuring the relevant parameters but also that the positioning cannot be separated from the challenges of space neutrality and printability of the sensors. From these challenges, further requirements for the positioning of the sensors can be derived. The position should be as close as possible to a location of (large) measurable deformation or in the vicinity of relevant loads, but it should be possible to realize it as neutral as possible to the installation space. Furthermore, the position and surface shape must be accessible for printing with different print heads. Moreover, the printed sensor must be contactable to transmit measurement signals. Finally, positioning the sensor in or near the load zone is another challenge.

An application of the sensors in the raceway, found on the inner surface of the outer ring as well as on the outer surface of the inner ring, would be a great challenge. In order to investigate its basic potential, this paper will first examine the deformation outside the raceway. In the case of the inner ring, the surface of interest would be the inner surface of the ring, which is more difficult to print on than an outwardly curved surface due to its inward curvature. As the sensor is pressed between the bearing ring and the housing wall in a real application (e.g., bearing installation in a gearbox), there are further challenges in terms of sensor damage and possible short circuits. Based on the advantages, disadvantages

and challenges mentioned above, the outer surface of the outer ring of a cylindrical roller bearing NU210 is chosen in this paper as the position for examining the potential of printed sensors for strain measurement on rolling bearings. Precisely, in this paper, the outer rib of the outer ring is first used to apply the sensor, and in the outlook (Section 5), the possibility of positioning the sensor on the outer surface of the outer ring is presented.

Another important requirement for the position is the material on which the sensor is printed. Many rolling bearings are made of 100Cr6 rolling bearing steel. Metallic surfaces are challenging for the application of printed sensors because they are electrically conductive. In order to print a strain gauge on the rings of a rolling bearing, the sensor must be insulated from the bearing surface. An alternative could be to use non-conductive components of the rolling bearing or non-metallic rolling bearings. A non-conductive component could be, for example, cages made of polymer, non-conductive rolling bearings, e.g., polymer bearings. On the other hand, cage deformations are not a relevant measurement parameter for mechanical stress, load determination and fatigue estimation of the bearing rings and are challenging due to the difficult accessibility and dynamic vibrations. Furthermore, the restriction to non-conductive rolling bearings for this general investigation of the potential of printed sensor technology for rolling bearings would not be useful at the beginning since the majority of rolling bearings are made of steel. Therefore, a rolling bearing ring made of 100Cr6 is considered in this paper, thereby creating the challenges of insulation, which will be discussed below.

### 2.2. Sensors Used and Printing Process

Figure 2 depicts the layout of the sensor structure with an active length of 5 mm that was to be printed onto the rolling bearing. In order to ensure electrical insulation between the sensor and the conductive surface of the bearing, an insulating coating of Genesink ProtectInk S [24] was applied manually with a squeegee onto the dedicated sensor area in the first step. The deposited coating was then cured in a Hönle UVA Cube 2000 with an iron radiator for 5 min at 1000 W.

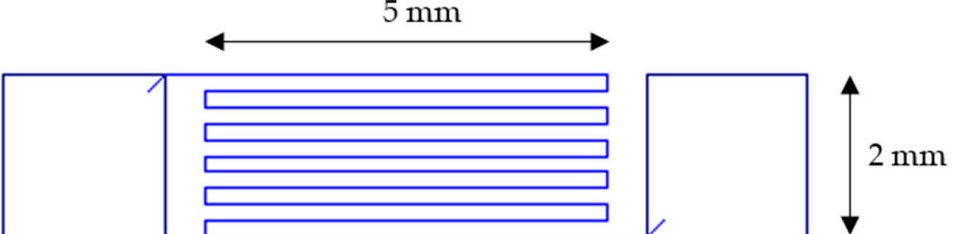

**Figure 2.** Layout and dimensions of the printed strain gauge.

In this work, the aerosol-based deposition system Nanojet by IDS [25] was employed to print the sensor structure. This process was based on creating a constant aerosol stream from a liquid ink containing functional particles in the nanometer range. Figure 3 depicts the print setup and shows the main components of the Nanojet printhead: The ultrasonic atomizer stimulated the ink in the reservoir and created an aerosol that was transported by the atomizer gas ($N_2$). In the flow cell, the aerosol was focused and accelerated by a surrounding sheath gas ($N_2$) before being deposited through the nozzle. A water-cooling system was used to dissipate the heat generated from the ultrasonic module and to ensure a constant process temperature. The Nanojet process decreased the generation of linewidths down to 20 μm. [26]

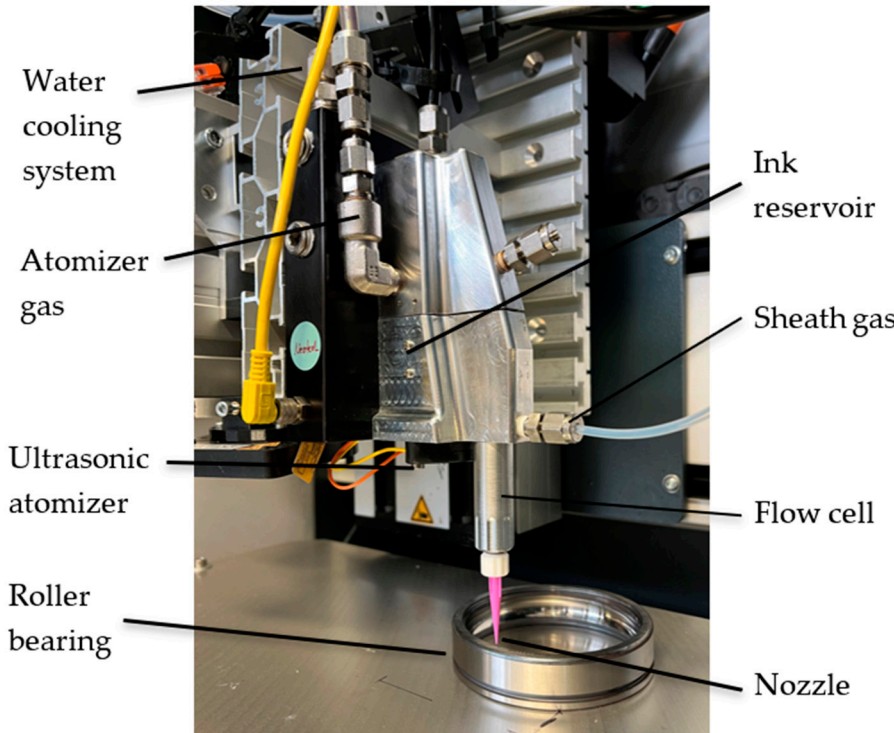

**Figure 3.** Setup for printing the strain gauge with Nanojet.

The printhead was mounted in a Neotech AMT 15X BT tabletop system [27] that controls the movement of the printhead via G-code. The sensor was printed with the conductive silver nanoparticle ink UTDots Ag40X [28]. For improved processability, 0.5 mL of the ink was diluted with 1.5 mL Xylene and 0.6 mL Terpineol. The utilized process parameters are shown in Table 1. In order to ensure continuous conductivity of the meander, the layout was printed three times at the same position. After printing, the structures were cured in a convection oven at 200 °C for 60 min. The printed sensor had a resistance of 170.1 Ω at a room temperature (RT) of 22 °C.

**Table 1.** Nanojet process parameters.

| Parameter | Value |
| --- | --- |
| Atomizer voltage | 27 V |
| Aerosol gas flow rate | 8 ccm |
| Sheath gas flow rate | 60 ccm |
| Chiller temperature | 22 °C |
| Lens diameter | 750 μm |
| Nozzle diameter | 150 μm |
| Printing velocity | 5 mm/s |
| Working distance | 2 mm |

In addition to the printed sensor, a conventional strain gauge 1LY13-6-350 [29] with a resistance of 350 Ω and an active length of 6 mm was manually applied using cyanoacrylate glue after an isopropanol pre-cleaning step to the same rolling bearing on the opposite side of the printed sensor. This sensor was used as a reference for later evaluation of the printed strain gauge.

*2.3. Experimental Setup*

As described in the introduction, the measured deformation values of the printed sensor were to be validated not only with an FE-simulation (see Section 2.4) but also with a classic glued-on foil strain gauge as a reference sensor. Therefore, the bearing ring in the experimental setup was equipped with two strain gauge sensors as shown in Figure 4 (reference sensor on the left and printed sensor on the right side).

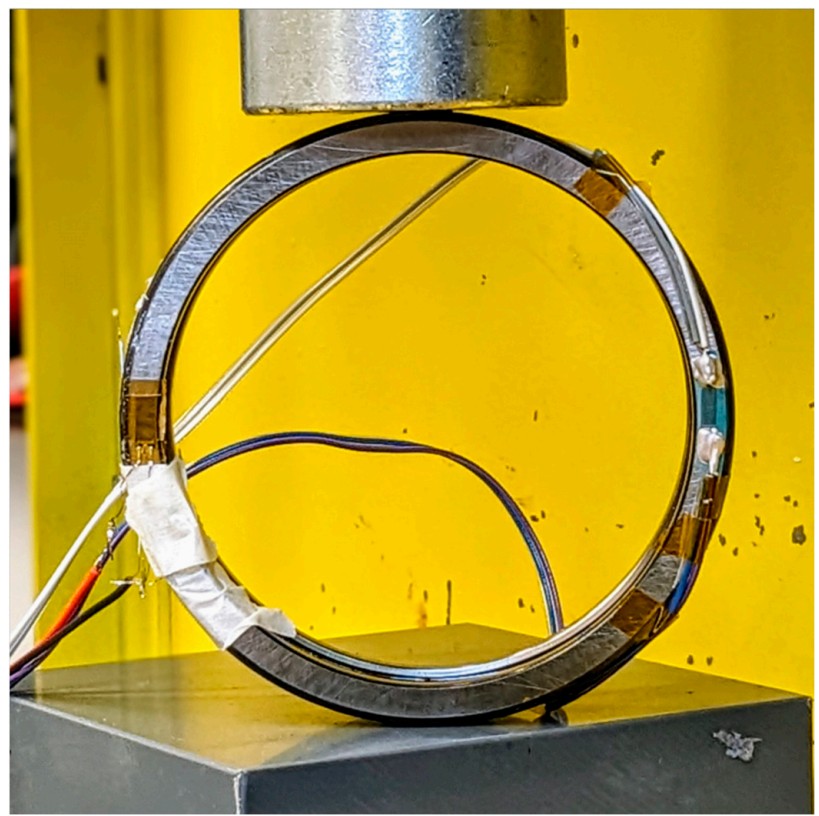

**Figure 4.** Outer ring with reference sensor (**left**) and printed sensor (**right**) (displacement sensor not shown).

The test setup consisted of a hand-operated hydraulic press with a tactile displacement transducer and two measuring devices for measuring the strain gauge sensors. The reference sensor was measured via a quarter-bridge circuit using HBM's QUANTUM X MX840A data acquisition system (see left side of Figure 5). The measurement output was the strain. The printed sensor was measured using a KETIHLEY DMM6500 laboratory multimeter in a four-wire-measurement setup, and the resistance was recorded as the measured value (see right side of Figure 5). The roller bearing outer ring (NU210) equipped with the sensors was manually clamped between two flat holders in the press, and the displacement transducer was zeroed. The outer ring was aligned so that the opposing sensors were positioned 90° to the direction of force application. The measurement was then carried out by compressing and relieving the outer ring by 2 mm in steps of 0.1 mm. After reaching the 2 mm compression, the outer ring was relieved in 0.5 mm steps. The entire compression and relief process was recorded with the two measuring devices. All measurements were taken at a room temperature (RT) of 22 °C.

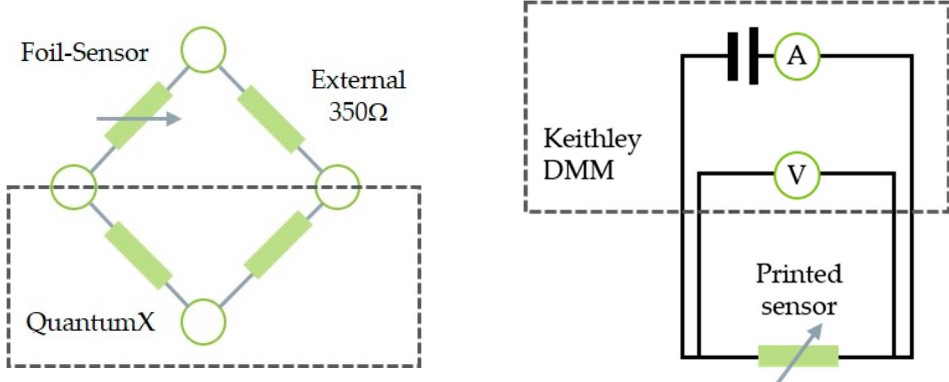

**Figure 5.** Electrical measurement arrangement from the reference sensor (**left**) and printed sensor (**right**).

### 2.4. Setup and Modeling of the FE-Simulation

To compare the experimental test results with theoretical expected values, an FEM model was built in parallel. The commercial FEM software ABAQUS 2021.HF7 was used.

Taking advantage of the symmetry, only a quarter of the rolling bearing outer ring was modeled, see Figure 6a. For this purpose, as shown in Figure 6b, the corresponding symmetry conditions were defined at the cross-sections of the XY and YZ planes. Based on the simplifying assumption that the XY-cross-section remained undeformed, the lower XY-cross-section was fixed. The load was applied in the form of a displacement of the upper XY-cross-section in the negative Y direction.

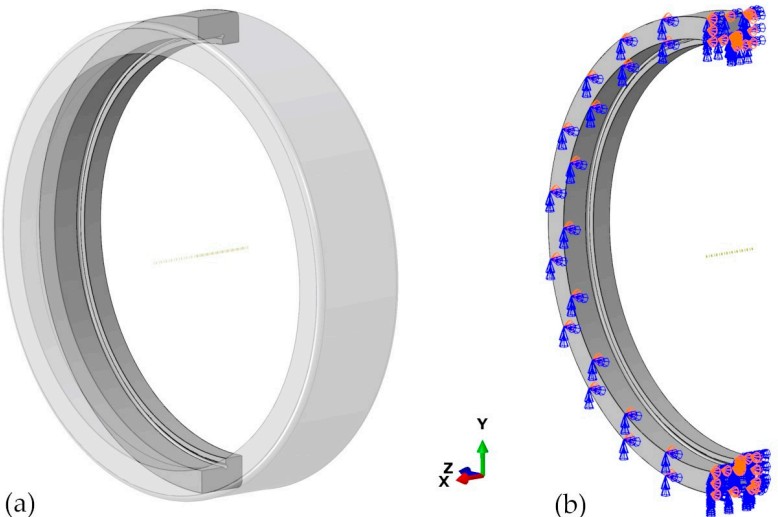

(a)                                                                    (b)

**Figure 6.** FEM model of the rolling bearing outer ring: (**a**) as a quarter model and (**b**) with boundary conditions.

Since all deformations remained in the purely elastic range, the definition of the rolling bearing steel 100Cr6 exhibiting purely elastic material behavior with an elastic modulus of 208,000 N/mm$^{-2}$ and a Poisson's ratio of 0.3 was justified. The geometry was meshed using C3D8R elements (8-node linear brick elements with reduced integration and hourglass control). For an enlarged section, see Figure 7. A relatively fine meshing with an approximate global size parameter of 0.25 was chosen in order to be able to determine the deformations, which were small in comparison to the overall dimension of the rolling bearing ring. The elements in the area of the center of the rolling bearing ring, which were evaluated, had a rectangular shape on the surface with edge lengths of 0.2 mm to 0.1 mm. Thus, the rectangular shape of the strain gauges could be easily taken into account during

the evaluation. This resulted in a total count of 170,376 elements. The displacement of 2 mm was built up through 20 steps and reduced to the initial state through 4 steps. The time sequence was selected in accordance with the experimental test sequence. The deformation, on the other hand, was defined idealized in 0.1 mm, respectively, and 0.5 mm steps.

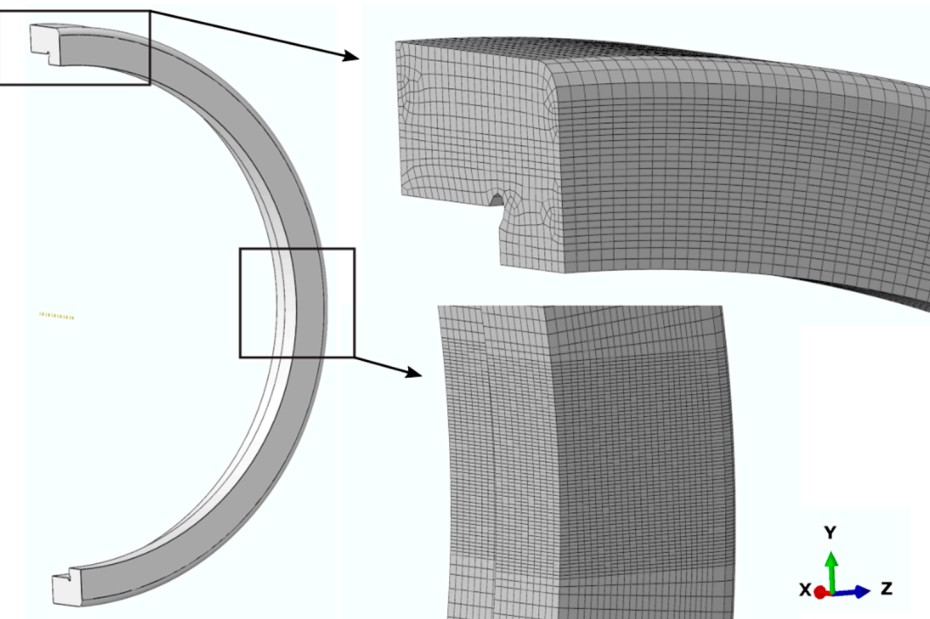

**Figure 7.** Enlarged sections of the meshing of the rolling bearing ring.

The evaluation of the FEM was closely related to the experimental mounting of the sensors and the measurement using the strain gauges. Since the two sensors attached to the rolling bearing ring measured an average strain in the vertical direction, the suitable average strain in the Y-direction (E22) of all surface elements corresponding to the area of the strain gauges in the test was determined in the FEM. This is illustrated in Figure 8.

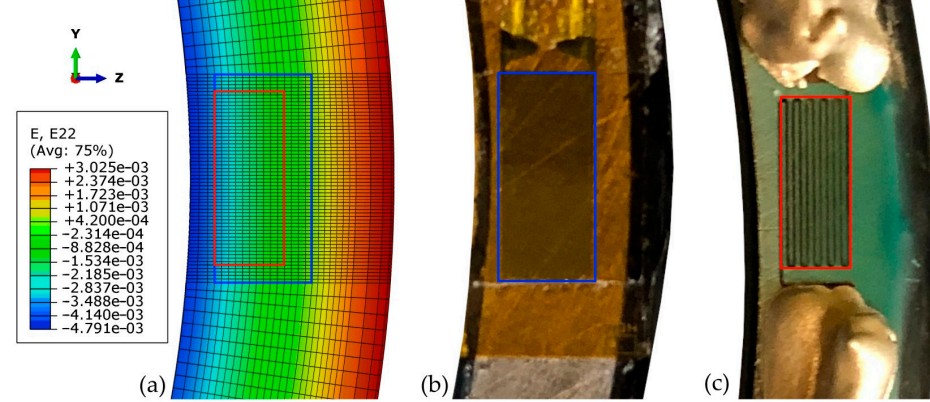

**Figure 8.** FEM result model with marked areas of the evaluated surface elements (**a**) and photos of the commercial reference (**b**) and the printed sensor (**c**) with equal scaling of the images.

## 3. Results

In the following Figures 9 and 10, the results of the measured strain of the reference sensor (1LY13-6-350) and the printed sensor (Ag Nanojet) are presented with the deformation calculated in the FE-simulation and are compared for validation.

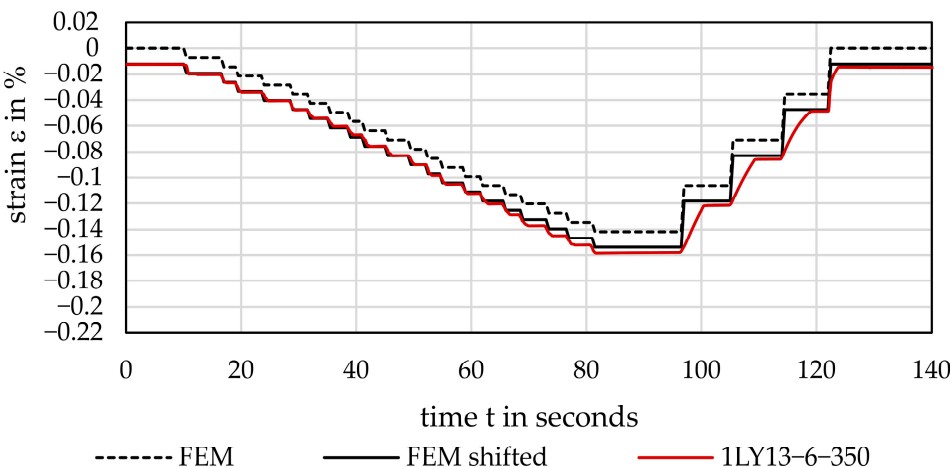

**Figure 9.** Comparison of the results of the strain measurement with the reference sensor 1LY13-6-350 (red line) with the FE-simulation (dashed and solid black lines).

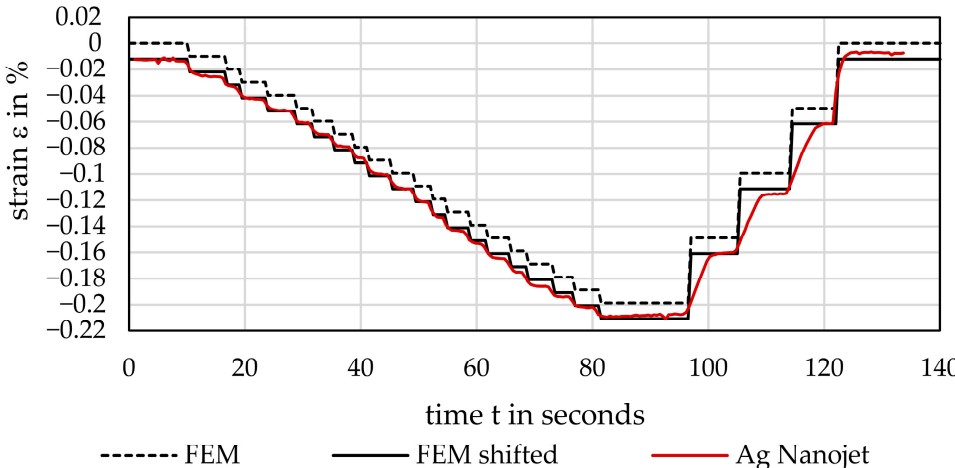

**Figure 10.** Comparison of the results of the strain measurement with the printed sensor Ag Nanojet (red line) with the FE-simulation (dashed and solid black lines).

### 3.1. Results of the Strain Measurement with the Reference Sensor and Validity with the FE-Simulation

In Figure 9, the plot of the measured strain of the reference sensor over time is shown with the red solid line. The curve of the measured strain shows a step-like and not a monotonously increasing or decreasing curve. The reason for the step-like curve is the manually controlled force application via the hydraulic press mentioned in Section 2.3. As mentioned above, a deformation of 0.1 mm is generated with each hand-operated pump stroke while compressing and of 0.5 mm while relieving.

The results of the FE-simulation to determine the strain at the reference sensor position are shown in Figure 9 with the dashed black line. The FE-simulation also takes into account the hand-controlled force application both in terms of deformation and over time. Therefore, the black line is also step-like and qualitatively matches the red line of the reference sensor but with an offset of 0.01% strain. The reason for the difference of 0.01% between the measured and calculated strain is that the ring was slightly preloaded by the weight of the hydraulic piston and for alignment prior to the measurement, then the installed dial indicator was calibrated to zero. In the FE-simulation, this preload was initially not taken into account, which explains the difference between the calculated and measured values.

To make the simulation results comparable with the measurement, the mentioned preload is added to the FE-simulation, and the results of this simulation are shown in

the black solid line in Figure 9. By taking the preload into account, there is a very good match between the measured and the simulated curves, especially in the compression period (time up to 80 s). In the relief period (time after 100 s), the results also coincide very well, but the curve looks slightly different due to the different manual force release of the hydraulic pressure.

Due to the good match between the simulation results and the measurement results with the reference sensor, the setting with the FE-simulation can be used to validate the measurement results of the printed sensor described in the following subsection.

### 3.2. Results of the Strain Measurement with the Printed Sensor

Figure 10 shows the comparison of the results of the strain measurement of the printed sensor (Ag Nanojet) with the results of the FE-simulation. In contrast to the commercial reference sensor, in the case of the printed sensor, the change in electrical resistance is measured during the deformation of the bearing ring. For this reason, a gauge factor (a correlation factor referred to as GF in the following) must first be determined for a printed sensor, with which the resistance signals are converted into a strain value.

In order to determine GF, it is necessary to initially calibrate the resistance signal once with real measured or calculated strain values. According to the validated measurement behavior of the reference sensor and the FE-simulation described above, the GF for the printed sensor can be calculated using

$$GF = \frac{R_{peak} - R_0}{R_0} \cdot \epsilon_{FEM} \tag{1}$$

whereby $\epsilon_{FEM}$ is the strain determined using the FEM, $R_0$ is the initial resistance value and $R_{peak}$ is the maximum value.

A simple determination of the GF based on the measurement results of the reference sensor has proven to be difficult. This is because the two sensors (reference and printed sensor) have a slightly different contact area with which they are attached to the bearing ring (see Figure 8). A strain gauge sensor always measures the average of all strains occurring under the contact surface, the so-called mean strain. If a strain gauge sensor is attached to a position of a mechanically stressed component in which a relatively homogeneous stress distribution prevails, the values of the strain at individual points under the contact surface of the sensor coincide with the average strain; the strain is thus independent of the contact surface. However, if the stress distribution is inhomogeneous, i.e., if there is a stress gradient, the average strain can no longer be used to determine the stresses at individual points under the contact surface. For the experimental setup as described in Section 2, the sensors are located at a position in the middle of the rib of the bearing ring. Figure 8 shows the stress distribution at this position. Due to the bending, there is a high stress gradient with a zero crossing (neutral fiber). For this reason, the measured values of the mean strain of the reference and the printed sensor must differ since they have a different contact area, and there is a gradient in the mechanical stress.

Due to the previously described circumstances, the FE-simulation is better suited for the determination of the GF, since the results of the FE-simulation take into account the respective areas of the sensors as described in Section 2.4. Therefore, values of the FE-simulation are used to determine GF. Through this fitting, a GF of 0.87 is determined.

Based on the determination of the gauge factor GF described above, the measured strain values of the printed sensor shown in Figure 10 are determined (red line, Ag Nanojet). As with the reference sensor, the FE-simulation of the strain (black dashed line) also takes into account the preload, and thus, the offset is calculated (black solid line). It can be seen that the measurement results of the strain of the printed sensor match very well with the expected calculated strains of the FE-simulation.

In conclusion, both strain measurements with the reference sensor and with the printed sensor provide valid values. The measured average strain of the reference sensor is

consistently smaller than that of the printed sensor, which is to be expected since it has a larger contact area and measures a larger unstrained area near the neutral fiber.

## 4. Discussion

For the printed strain sensor, the sensor response follows the strain load with a small delay and less steep falling and rising edges compared to the commercial reference sensor. This is attributed to the dielectric layer between the sensor and the bearing, which mediates the strain to the sensor itself and delays the signal by temporarily damping the deformation of the metal. There is also a slight overestimation of the strain after relaxation, likely due to the fact that the resistance values are measured without temperature compensation. Since printed silver structures have a significantly higher temperature coefficient of electrical resistance of about 0.276%/°C [30] than constantan alloys in foil strain gauges with about 0.004%/°C, such deviations of the baseline occur as soon as only the resistance without bridge circuit/T-compensation is measured. This influence is eliminated via software. The GF is temperature-dependent and is determined by the materials used, the adhesives and the match between the measuring grid and measuring object. For foil strain gauges with PI substrate and cyanoacrylate, the GF increases by approx. 0.6% [31]

The GF for printed silver-based strain gauges is also temperature-dependent. As recent results show [30], the GF change between 20 °C and 60 °C is significantly higher for silver strain gauges than for foil strain gauges, with an increase of 20%. In [30], the temperature sensitivity was also investigated. The silver gauges were stable in the range −40 °C to +100 °C after temperature cycling for 148 h.

The feasibility of strain sensing was shown, and the steps in the response signal upon a change in load are clearly visible. It is noted that the sensor signal can be used to estimate the load condition on the bearing ring. This is confirmed with the FEM analysis comparing the expected signal to the actual load of the reference and the printed sensor. Deviations arise from the fact that the bearing was preloaded before the 2 mm displacement measurement was started (see section above), explaining the vertical shift of the fitted FEM calculated curve.

In general, the precise positioning of the sensor through the printing process is an advantage. However, there are challenges for applications on bearings for precise strain measurement, partly by means of realizing adequate insulation of the conductive surfaces with as minimal influence on the sensor response as possible. Moreover, the inertia of the sensor response might impact the achievable read-out frequencies for dynamic applications but could possibly be mitigated by prolonged sintering or higher temperatures during the sintering process to form a more bulk like material that exhibits less creep in the structure. Additionally, ultra-thin insulation layers can lead to a quicker response of the sensor.

The experimentally determined gauge factor for compression of the printed sensor $GF_{printed} = 0.87$ is only an approximation. The position of the printed sensor is not completely congruent with that of the reference sensor. The overall dimensions differ slightly, and more importantly, the radial position also deviates from that of the reference position. Hence, from the FEM analysis, it is derived that the printed sensor covers less of the neutral strain area, i.e., without significant load inside the bearing wall. Furthermore, there is no uni-axial strain condition, which is required to define the gauge factor GF precisely, since it is normally determined on tension rods. Instead, due to the curved surface, multi-axial strain conditions in radial direction occur, effectively reducing the resistance response of the printed sensor.

Cross-checking the pull-strain response of the printed sensor yields a GF of approx. 1.8, which is in the range of previously published values on tension rods [30].

## 5. Conclusions

The goal of this paper was to demonstrate a validated application of printed sensors as strain gauges on a rolling bearing under isolated boundary conditions (i.e., not in real-

world operation) to illustrate the general feasibility and to identify potential problems and challenges that may arise during implementation.

The results showed that a printed sensor on a rolling bearing ring can be used to validly measure the deformation. During the manufacturing and printing process, it was possible to apply the conductive tracks, including an insulation on the rolling bearing steel. The electrical contacting was also successful. This provided answers to the questions raised by the above-mentioned challenges and carried out preliminary work for the use of printed sensor technology under real operating conditions.

For the measurement position, the outer rib of the outer ring of the rolling bearing was chosen. This position has proven to be advantageous for printing, insulation and contacting. On the other hand, this advantage in relation to the printing position resulted in challenges to determine the necessary GF with the use of a reference sensor, due to the large stress gradient on the rib, and the comparability between the printed and reference sensor strongly depends on the contact area. For this reason, the GF was determined by means of the FE-simulation, which led to valid results.

Thus, a proof of concept for printed sensors on rolling bearings was demonstrated, but the chosen position proved to be disadvantageous with sensors of different contact areas at the same time. In order to improve this, there are two possibilities, which can also be used in combination: 1. use of a reference sensor with the same area, to compare the average strains and determine the GF; 2. positioning of the strain measurement at a position of the rolling bearing where a homogeneous strain can be expected.

As described above, the choice of the position of the printed sensor depends, on one hand, on whether a relevant measured variable (strain, load, etc.) is to be measured on the rolling bearing during the operation, and on the other hand, on whether this position can be printed properly. As described in Section 2.1, the outer rib of the rolling bearing ring was chosen among other things on account of its flat surface, which is well suited for printing. The printing, insulation and contacting process on curved surfaces is more challenging and should be investigated in further work. Figure 11 shows our first, more advanced attempt to enable strain measurements on the outside of the rolling bearing ring using a printed sensor.

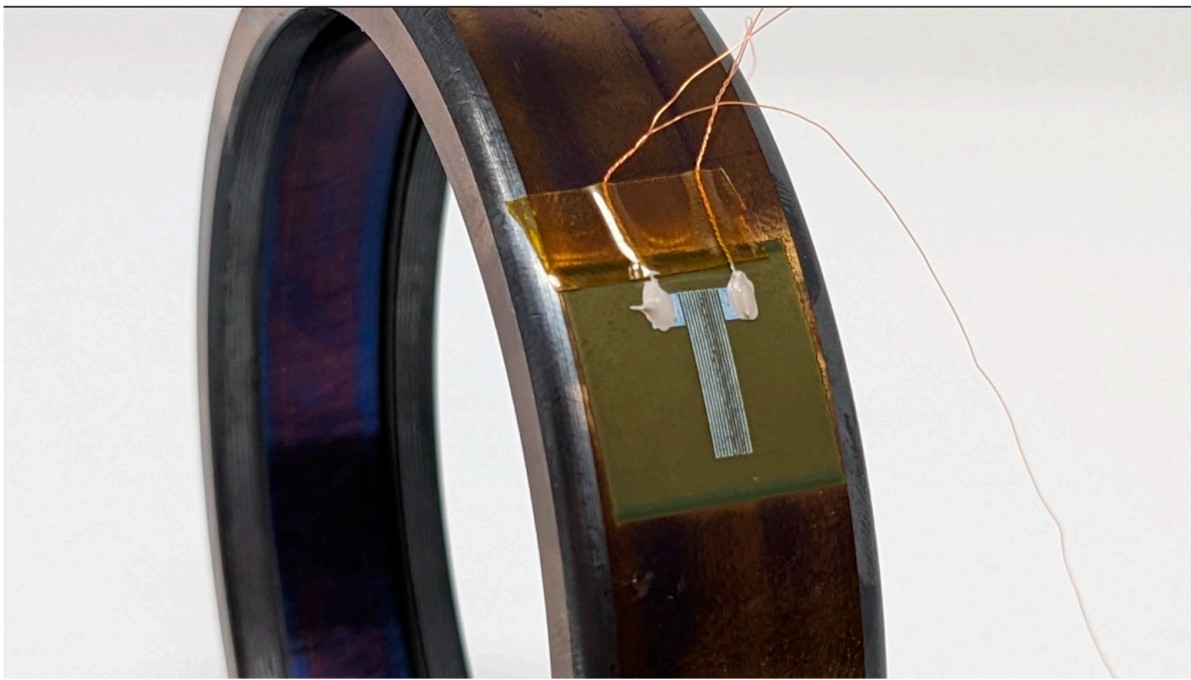

**Figure 11.** Attachment of a printed sensor on the outside of a rolling bearing ring and the challenges of insulation and contacting when the surface is curved.

The position on the outer surface has some advantages, e.g., a larger available area for printing than on the rolling bearing rib and also a more homogeneous expected (averaged) strain, which is better for determining GF using a reference sensor. Although the printing process has been shown to work, the challenge of the curved surface has so far made insulation and contacting difficult, resulting in unwanted short circuits. There is a need for further research here because if the contacting and insulation are successful, other interesting, curved surfaces on the rolling bearing (including the inner ring) can be investigated in order to measure relevant strains in the long term, also in operating conditions, and to derive conclusions about loads and load distribution. There is also potential to measure strains close to the load zone or the raceway. However, further research is required in order to ensure the neutrality of the installation space and to avoid damage to the sensor during operation.

According to the findings in this paper, the gauge factor GF would always have to be validated once at the beginning of each further application or once again when the geometry or position is changed. This would be completed using a validated FE-simulation, as it was conducted in this paper. Alternatively, this could be completed using a reference sensor that is expected to have the same mean strain as the printed sensor. According to the findings from the work presented in this paper, GF needs to be determined only once, but it needs to be determined once again for a different position or when the geometry is changed. There is a need for further research regarding the influence of the printing on GF. Other operating conditions—such as temperature influences, etc.—must also be investigated in future work.

Moreover, long-term tests with constant and alternating loads should be carried out by means of test rig investigations in order to investigate the influences of real operating conditions on the printed sensor, e.g., on durability and on GF. Thereby, the main advantages of printed strain gauges compared to conventional gauges can be investigated, such as space neutrality, self-supply via energy harvesting and the installation and acquisition of measured data at positions that are not possible for conventional sensors.

In addition, further work should also focus on integration into functional surfaces of the rolling bearing and good suitable measurement surfaces. Finally, specific implementation in real applications should be carried out to measure real operating data, e.g., in the context of condition monitoring and predictive maintenance processes. Possible applications include large-diameter bearings and slewing rings in wind turbines.

**Author Contributions:** Conceptualization, M.B., F.H. (Fabian Halmos), M.A., F.H. (Felix Häußler), M.J. and S.W. (Sven Wirsching); methodology, M.B., F.H. (Fabian Halmos), M.A., F.H. (Felix Häußler) and M.J.; measurement, F.H. (Fabian Halmos), F.H. (Felix Häußler), M.J., M.A. and S.W. (Sven Wirsching); simulation, M.J.; validation, M.J., F.H. (Fabian Halmos), M.A. and F.H. (Felix Häußler); printing of sensors, M.A.; sensor measurement consulting, J.R.; writing—original draft preparation, M.B.; writing—review and editing, M.B.; visualization, F.H. (Fabian Halmos), M.A., F.H. (Felix Häußler) and M.J.; supervision, M.B., M.R., J.F. and S.W. (Sandro Wartzack). All authors have read and agreed to the published version of the manuscript.

**Funding:** The printing and measuring were partly supported in the course of the project PreSenS (21173N) funded by the German Ministry of Economic Affairs and Climate Action through AiF-IGF research association 3D-MID e.V.

**Data Availability Statement:** For insight into the measured values and the FE-simulation, please contact the corresponding author.

**Acknowledgments:** We acknowledge the support from Friedrich-Alexander-Universität Erlangen-Nürnberg and Technische Hochschule Nürnberg Georg Simon Ohm.

**Conflicts of Interest:** The authors declare no conflict of interest.

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
