# Peer review of "Use of Printed Sensors to Measure Strain in Rolling Bearings under Isolated Boundary Conditions"

_lubricants, doi:10.3390/lubricants11100424_

Round 1

Reviewer 1 Report

The manuscript investigates the potential of using aerosol-based method to print deformation sensors in rolling bearings. The assessment is done under static conditions with a radial load applied to a bearing outer ring. Then the measured deformations are compared with FE analysis. It is concluded that there is good potential of this technology to be used in real applications. However, selection of an optimal position for the sensors and the proof of the technology in real working environment in a bearing remains to be demonstrated.

The reviewer is favorable to the publication of this manuscript with the following major revisions/answer-to-questions.

1. Once the sensors are printed, what are the effects of high and low temperature on the sensor? Why did the authors omit experiments with different temperatures? What are the temperature limits in operation for this technology ? Will the GF change with temperature?

2. What are the next steps to overcome the challenges that the authors foresee to bring this technology to real bearing applications? Can the authors elaborate in the discussion section ?

3. Can the authors comment on potential vibration effects on the sensor and its measurements?

Reviewer 2 Report

The paper presents a case study of a printed strain sensor to the deformation analysis of rolling bearings. Even though the level of innovation of the paper is moderate, the research methodology is sound, and the results are very well described. I only have minor comments on the paper:

-  There are typos (repeated words) at line 103 and 400.

-  Line 219-220: which temperature is reached during the curing procedure? If known, this datum may be of interest for the reader.

- In the conclusion section, it seems to me that the main message is that the directly printed sensors can be applied to the case under examination. What remains somehow hidden is: what are the advantages, if any, compared to the use of conventional strain gauges that emerged from the experiments? I believe that it would be useful to spend a few more words on that in the conclusion.
